# Energy Imbalance Gap, Anthropometric Measures, Lifestyle, and Sociodemographic Correlates in Latin American Adults—Results from the ELANS Study

**DOI:** 10.3390/ijerph19031129

**Published:** 2022-01-20

**Authors:** Martha Cecilia Yépez García, Marianella Herrera-Cuenca, Gerson Ferrari, Lilia Yadira Cortés Sanabria, Pablo Hernández, Rafaela Yépez Almeida, Mónica Villar Cáceres, Georgina Gómez, Rossina Pareja, Attilio Rigotti, Irina Kovalskys, Mauro Fisberg

**Affiliations:** 1Colegio de Ciencias de la Salud, Universidad San Francisco de Quito, Quito 17-1200-841, Ecuador; rafaelayepeza7@gmail.com (R.Y.A.); mvillar@usfq.edu.ec (M.V.C.); 2Centro de Estudios del Desarrollo, Universidad Central de Venezuela (CENDES-UCV), Caracas 1053, Venezuela; marianella.herrera@ucv.ve; 3Fundación Bengoa, Caracas 1053, Venezuela; 4Escuela de Ciencias de la Actividad Física, el Deporte y la Salud, Universidad de Santiago de Chile (USACH), Santiago 7500618, Chile; gerson.demoraes@usach.cl; 5Departamento de Nutrición y Bioquímica, Pontificia Universidad Javeriana, Bogotá 110231, Colombia; ycortes@javeriana.edu.co; 6Escuela de Nutrición y Dietética, Facultad de Medicina, Universidad Central de Venezuela, Caracas 1053, Venezuela; doctuscumliber@gmail.com; 7Departamento de Bioquímica, Escuela de Medicina, Universidad de Costa Rica, San José 11501-2060, Costa Rica; georgina.gomez@ucr.ac.cr; 8Instituto de Investigación Nutricional, La Molina, Lima 15026, Peru; rpareja@iin.sld.pe; 9Centro de Nutrición Molecular y Enfermedades Crónicas, Departamento de Nutrición, Diabetes y Metabolismo, Escuela de Medicina, Pontificia Universidad Católica, Santiago 8330024, Chile; arigotti@med.puc.cl; 10Carrera de Nutrición, Facultad de Ciencias Médicas, Pontificia Universidad Católica Argentina, Buenos Aires C1107 AAZ, Argentina; ikovalskys@gmail.com; 11Centro de Excelencia em Nutrição e Dificuldades Alimentaes (CENDA), Instituto Pensi, Fundação José Luiz Egydio Setubal, Hospital Infantil Sabará, São Paulo 01228-200, Brazil; mauro.fisberg@gmail.com; 12Departamento de Pediatria, Universidade Federal de São Paulo, São Paulo 04023-061, Brazil

**Keywords:** energy intake, energy expenditure, energy balance

## Abstract

Overweight and obesity are often explained by an imbalance between energy intake and expenditure. This, in addition to metabolic effects, makes it difficult to assess the real state of individual energy balance. This study aims to analyze the energy gaps between intake and expenditure in the adult population of Latin America, as well as its relationships with sociodemographic variables and nutrition status, to draw an epidemiological perspective based on the trends observed. The energy imbalance gap was used to this end. The difference between energy intake and expenditure can be applied as a reference to explain whether weight equilibrium can prevent weight gain. Moreover, the energy imbalance gap allows for a better understanding of the design of public health policies. Using data from the Latin American Study of Nutrition and Health, the energy imbalance gap in adult population from eight Latin-American countries was assessed in 5994 subjects aged from 19–65. Usual dietary intake was measured using two non-consecutive 24 h dietary recalls. The sociodemographic questionnaire was supplemented by anthropometric measurements. Physical activity was measured through the long International Physical Activity Questionnaire. Energy expenditure was obtained using the basal metabolic rate. For the overall sample, the mean energy intake was 1939.1 kcal (95% CI: 1926.9; 1951.3), the mean of energy expenditure was 1915.7 kcal (95% CI: 1906.4; 1924.9), and the mean of energy imbalance gap was 23.4 kcal (95% CI: 11.9; 35.0). Results show that energy intake and expenditure were higher in men. Moreover, subjects aged 19–34, of high socioeconomic level, who completed high school, were mestizos and were of normal weight consumed the highest number of calories. Overall, a positive energy imbalance gap was observed. Overweight and obese from Argentina, Costa Rica, Ecuador, Peru, and Venezuela showed a significantly lower energy imbalance gap than underweight subjects. These findings confirm the high variability of energy imbalance gap and the accompanying correlates of energy intake and expenditure. Further research is needed to specifically address interventions in low and middle-income countries such as many in Latin America, to help reduce the prevalence of obesity and eradicate undernutrition.

## 1. Introduction

Progress in understanding how body weight is regulated has been achieved in recent decades. Particularly, overweight and obesity are frequently explained by an imbalance between energy intake and expenditure. For this oversimplified perspective, some components should be considered, such as the food people consume or their daily physical activity management [1,2]. However, what a person eats, as well as the physical activity achieved, reflects a lifestyle conditioned by the environment where an individual lives (personal security to go outdoors or going to work by walking, which can condition the level of physical activity or sedentary habits) [1,3,4], access to quality food, food behavior, temperature, basal metabolic rate, and social isolation and stay at home messages, to mention a few [2,5]. In the end, these factors have an impact on the health of individuals for whom energy balance management is key to maintaining good nutritional status [2,5]. The American Society of Nutrition and other allied organizations proposes a three-component energy balance model for assessing energy balance, as follows: components of intake, components of expenditure, and components of storage [6,7]. When energy intake and energy expenditure are equal, the body’s energy balance (EB) is stable. When energy intake is excessive, the EB is positive, and when the energy expenditure is larger than the intake, EB is negative [4].

The physiological mechanisms for controlling the EB are still under study, yet the evidence suggests complex regulation involving neuro-endocrine-gastrointestinal signaling pathways through which appetite and satiety would not be as variable over time, deriving in body composition according to the respective energy intake and expenditure [1,2].

The components of EB, namely energy intake, expenditure and storage, translate into energy intake in the form of fats, protein, carbohydrates and alcohol; energy expenditure through resting metabolic rate (RMR), the thermic effect of food (TEF) and the energy expended through physical activity, which varies by activity type and duration [1,3].

Some other factors can modify the risks for weight gain. For instance, breastfeeding during early infancy and later high dietary fiber intake and consumption of fruits and vegetables decrease overweightness and/or obesity risks. In turn, excessive intake of energy-dense foods and drinks increases these risks [8,9,10,11].

If these components, energy intake and energy expenditure, were to remain only on its biological parameters, it would be easier to assess and have a complete understanding of human EB. Nevertheless, when addressing EB components, the evidence states that environmental influences energy intake or expenditure [2,11]. Furthermore, the wide variability on intake and expenditure and the metabolic effects, even within the same individual [2,11,12], hamper the assessment of the real state of subject’s energy balance. This also complicates addressing populations since ranges for energy expenditure and energy intake equilibrium are not established. Food availability, economic scenarios, and social changes determine what people eat and the quality of this, in the same way that the type of work, and the place where people live, still influence the means people use for commuting.

In the literature reviewed, some research studies on different populations address EB, particularly to understand the global trend toward obesity [13,14,15,16]. Additionally, the term ‘energy imbalance gap’ has been introduced to better understand for public health policies, in which the difference between energy intake and expenditure can used a reference to establish whether weight can reach equilibrium and prevent weight gain [17,18]. Even when traditional methods for addressing energy balance and metabolic rates, such as indirect calorimetry or doubly labeled water, are more accurate procedures to measure this indicator, both are costly and complex to implement in public health. Thus, the energy imbalance gap in the Latin American Survey of Nutrition and Health (Estudio Latinoamericano de Nutrición y Salud, ELANS) population was explored by addressing the differences between energy intake and expenditure to have an epidemiologic perspective on the trend. Few studies have been conducted in Latin America. This paper aims to study the gaps between intake and expenditure in the adult population in Latin American countries, and its relationships with sociodemographic variables and nutrition status.

## 2. Materials and Methods

### 2.1. Study Design and Sample

ELANS is a household-based multi-national and cross-sectional study carried out in order to collect reliable and comparable information about energy intake and energy expenditure in representative samples of eight Latin American countries (Argentina, Brazil, Chile, Colombia, Costa Rica, Ecuador, Peru and Venezuela). The data used in this study was gathered between September 2014 and July 2015 [19,20].

The ELANS study enrolled 10,134 subjects between 15 to 65 years of age, including 5994 adults aged 19 to 65 (Figure 1), using a randomized complex multistage sampling process stratified by geographical location, sex, age, and socioeconomic status (SES), with a random selection of primary and secondary sampling units for the urban population in order to achieve a representative sample [19]. Sample size was calculated using a confidence level of 95% and a sample error of 3·49% at a 5% significance level; in addition, a survey design effect of 1.75 was estimated based on the guidelines of the US National Center for Health Statistics [21]. The overall rationale, study design and standardization of the food composition database are described elsewhere [19,20]. To participate in the study, all participants signed an informed consent. The Western Institutional Review Board approved the study protocol (# 20140605), which was registered at Clinical Trials (#NCT02226627).

Participants who had significant physical/mental impairment that impacted food intake and physical activity levels, pregnant or lactating women, and aged <15 or >65 were excluded. Adolescents (15 to 18 years old) were excluded from the analyses as well because the ELANS study did not include adolescents of all ages. In addition, adolescents may have restricted independent mobility [22] that may yield physical activity associations different from those observed in adults. Furthermore, physical activity guidelines for adolescents differ from those for adults [23].

### 2.2. Sociodemographic Variables

A sociodemographic questionnaire was taken to collect information on basic demographics and socioeconomic variables. The variables consider the following: sex (male and female); age group (younger adults (19–34 years), adults (35–49 years) and older adults (50–65 years)); SES (low, medium, or high); education level (none and primary, high school, or bachelor’s degree); and race/ethnicity (Caucasian, mestizo, black, indigenous, other) [19].

### 2.3. Anthropometry

The anthropometric variables considered for this study were body weight and height, which were calculated using standard procedures and equipment. Bodyweight (kg) was measured with calibrated electronic scales (Seca^®^, Hamburg, Germany), with an accuracy of 0.1 kg. Body height (cm) was measured with a portable stadiometer with an accuracy of 0.1 cm.

The body mass index (BMI) was calculated from height and weight and interpreted using the international classification of the BMI of the World Health Organization (WHO): underweight <18.5; normal 18.5–24.99; overweight ≥ 25.0–29.99; and obesity ≥ 30.0 [24].

### 2.4. Dietary Intake

Dietary intake was assessed using two 24 h food recalls applied on two nonconsecutive household visits, including weekdays and weekend days. A photographic album of common foods of each country and household utensils were used to estimate portion sizes, while the Multiple Pass Method was employed to assess all foods and beverages consumed over the previous day [25,26]. The food intake information obtained was transformed into volumetric measurements (grams and milliliters) by critical nutritionists trained for this activity.

The data obtained from macronutrients on the dietary recall questionnaires were converted into energy using the Nutrition Data System for Research (NDS-R) [26]. The web-based statistical modelling technique Multiple Source Method (MSM), proposed by the European Prospective Investigation into Cancer and Nutrition (EPIC), was used to estimate the usual energy intake considering within-person variance [27]. Misreporting was included as an adjustment [28].

### 2.5. Physical Activity

The International Physical Activity Questionnaire (IPAQ) validated for Latin America was used in its extended Spanish version and localized to the everyday language of each participating country [29,30,31]. The domains of transport and leisure-time physical activity were included. Information was collected during the second household visit. Details on the development, reliability, and validity of the IPAQ are available elsewhere [24]. Only the active transportation and leisure-time physical activity sections were included due to the greater relevance of these domains for guiding public health policies and programs [29] and the relatively low validity of the IPAQ items on occupational and home-based physical activity questions in Latin American urban settings.

Thus, data on physical activity from the questionnaire was reported as minutes/day of walking, moderate and vigorous physical activity. The metabolic equivalents (METs)—by minutes/day and minutes/week (MET–min/day and MET–min/week, respectively)—in each physical activity were calculated according to the Compendium of Physical Activities [32,33]. Data were analyzed following the IPAQ scoring protocol (https://sites.google.com/site/theipaq/scoring-protocol, accessed on 18 January 2022), and the participants were divided into the following 3 groups:(1)A high group that included participants who had performed vigorous-intensity activity for at least 3 days and accumulated at least 1500 MET–min/week, or any combination of walking, moderate-intensity, or vigorous-intensity activities for 7 days, achieving a minimum of 3000 MET–min/week.(2)A moderate group which included participants who had performed vigorous activity for 3 or more days, at least 20 min per day, or ≥5 days of moderate-intensity activity or walking, at least 30 min per day, or ≥5 days of any combination of walking, moderate-intensity, or vigorous-intensity activities, achieving a minimum of 600 MET–min/week.(3)A low group which included participants who had not met any of the above recommendations.

### 2.6. Energy Expenditure

Total energy expenditure (EE) was obtained using the basal metabolic rate (BMR) proposed by FAO/WHO [34] and the activity factors published by Gerrior, S. et al. [35], using the following formula [34]:*Energy Expenditure Equation: (EE = BMR xPA)*

The equations used for BMR are the ones presented on the “Human Energy Requirements” report [34], which comprise sex, age and weight, as shown below.
AgeMaleFemale18–3015,057 * kg + 692·214,818 * kg + 486·630–6011,472 * kg + 873·18126 * kg + 845·6≥6011,711 * kg + 587·79082 * kg + 658·5

The activity factors used consider sex and physical activity levels (low, moderate and high) shown below [35], data obtained with the extended version of IPAQ [31].
PALMale PAFemale PALow1.121.14Moderate1.271.27High1.541.45

### 2.7. Energy Imbalance Gap

The energy imbalance gap (EIBG) was used as a method to have a relationship between EI and EE because the referred methodologies for energy balance have elevated costs, are cumbersome and cannot be performed in public health studies such as ELANS. Therefore, EIBG was calculated using the difference between EI and EE (EI-EE = EB). Negative values showed that people waste more energy than they intake, and positive values showed that people have a higher intake than expenditure [36].

### 2.8. Statistical Analysis

Statistical analyses were carried out using the software IBM SPSS (V26, SPSS Inc., IBM Corp., Armonk, New York, NY, USA). The Kolmogorov–Smirnov test was used to verify if data were normally distributed. Means, 95% confidence interval (95% CI), specific percentiles (3rd, 10th, 25th, 50th, 75th, 90th and 97th), and percentages were computed, as needed, to describe the variables. Weighting was conducted according to sociodemographic characteristics, sex, socioeconomic level, and country.

Multilevel linear regression models were used to examine the associations between sociodemographic characteristics (independent variables) with energy intake, energy expenditure, and energy imbalance gap (dependent variable) for each country and overall. The models included region and cities as random effects. Moreover, they were adjusted for sex, age group, ethnicity, socioeconomic level, education level, ethnicity, and body mass index, as well as reported unstandardized beta coefficients and 95% CI. A significance level of 5% was adopted.

## 3. Results

The sample characteristics are shown in Table 1. Overall, 5994 adults aged 19–65 completed the questionnaire. For the total sample, mean energy intake was 1939.1 kcal (95% CI: 1926.9; 1951.3), mean of energy expenditure was 1915.7 kcal (95% CI: 1906.4; 1924.9), and mean EIGB was 23.4 kcal (95% CI: 11.9; 35.0).

Regarding energy intake, Chile had the lowest values (mean 1767.0 kcal; 95% CI: 1730.3; 1803.7), and Ecuador showed the highest average (mean 2128.3 kcal; 95% CI: 2089.1; 2167.1); the difference between these two countries was 361.1 kcal (Table 1 and Figure 2A). This is similar to the percentiles analysis of energy intake showed in Figure 2A. In general, mean energy intake was lower for women compared to men. In addition, the age group between 19 and 34 years old, high socioeconomic status, high school education level, mestizos, and subjects with normal weight consumed the highest number of calories (Table 1).

For energy expenditure, Ecuador was the country that spent the largest number of calories (mean 2005.2 kcal; 95% CI: 1972.1; 2038.2) and Colombia was the one that spent the lowest (mean 1884.2; 95% CI: 1859.1; 1909.3), with a mean difference of 121 kcal between these two countries (Table 1 and Figure 2B). Percentiles are also presented in Figure 2B. In addition, men spent more energy than women, as well as subjects between 19 and 34 years of age, high socioeconomic status, high school educational level, black ethnicity, and subjects with obesity (Table 1).

For EIBG, Figure 2C reported that Peru had the highest positive EIBG at percentile 50th and Chile the highest negative EIBG (142.6 kcal and −199.9 kcal, respectively). As Table 1 shows, men had a negative EIBG while women had a positive one. In addition, subjects between 19 and 34 years old had a positive EIBG compared to the rest of age groups, as well as low socioeconomic status, mestizos, and underweight subjects. However, the analysis of the end percentiles demonstrated that at 75th percentile, the overweight population had an average that increases at 301.03 kcal, whereas the obese population had EIBG of 154.10 kcal (percentiles analysis of EI, EE and EB for all variables are presented as Appendix A).

The linear regression model showed that an increase of one point on the scale of the independent variable was associated with a proportional increase or decrease in beta coefficients of energy intake, energy expenditure and EIBG (Table 2, Table 3 and Table 4). In the overall sample, the results showed that men (β: 402.8; 95% CI: 380.9; 424.7), aged 19 to 34 (β: 200.5; 95% CI: 171.3; 229.8) and 35 to 49 years (β: −118.6; 95% CI: 89.8; 147.4) and those obese (β: 111.4; 95% CI: 28.6; 194.2) had higher energy consumption compared to women, older adults (50 to 65), and underweight, respectively.

In addition, it was observed that men consumed more energy compared to women in all countries, as well as subjects under 49 years of age, and except for subjects between 35 and 49 years of age from Argentina and Venezuela (Table 2).

In Table 3, where energy expenditure was compared with the sociodemographic variables, it can be observed that, overall, men (β: 469.4; 95% CI: 457.0; 481.8), participants between 19 and 34 years old (β: 217.3; 95% CI: 200.1; 234.4) and 35 to 49 years old (β: 110.9; 95% CI: 95.5; 126.2), those of medium (β: 28.7; 95% CI: 15.3; 42.2) and high socioeconomic status (β: 53.9; 95% CI: 29.8; 78.0), and participants with normal weight (β: 180.3; 95% CI: 135.6; 224.9), overweight (β: 342.8; 95% CI: 298.8; 386.8) and obesity (β: 550.9; 95% CI: 502.3; 599.5) spent more energy compared to women, older adults, people with low socioeconomic status and subjects who were underweight, respectively. Furthermore, in Brazil, subjects with high school studies and all ethnic groups were observed to spend more calories than subjects with primary education and indigenous, respectively.

The association between correlates and total energy imbalance gap (Table 4) showed that men (β: −66.6; 95% CI: −89·1; −44.1) had a higher negative EIBG compared to women, and those with normal weight (β: −144.7; 95% CI: −224.5; −64.8), overweight (β: −285.5 95% CI: −369.5; −201.5) and obesity (β: −439.5; 95% CI: −525.5; −353.6) had lower EIBG than underweight women, respectively. In Brazil, Colombia, Ecuador and Venezuela, men had a higher negative balance compared to women. In Venezuela, subjects with a medium and high socioeconomic status had a positive EIBG compared to the lower socioeconomic status. Lastly, in Argentina, Costa Rica, Ecuador, Peru, and Venezuela, the EIBG was significantly lower in overweight and obese subjects compared to underweight subjects.

## 4. Discussion

Energy balance and its components are complex and important indicators for understanding the nutritional status of different age groups and its characteristics. Energy balance refers to the fact that equilibrium will be achieved when the intake of foods equals the energy expenditure of an individual. Measuring those can be challenging as methods of doubly labeled water [37,38] and indirect calorimetry cannot be used in household surveys such as this study [39]. However, knowing the gaps between energy intake and expenditure is key to understand where the population stands at the moment of this cross-sectional study.

Obesity occurs when a positive energy balance is sustained over time [40,41], and the interacting factors that influence energy intake and physical activity related to the environment and other biological parameters express on the population’s lack of equilibrium for maintaining a healthy weight [4,40,42,43,44]. In turn, negative energy balance, predominantly by energy deficiencies, leads to progressive weight loss with new energy expenditure patterns to adapt into a new lower level of equilibrium [34,45].

In this study, EIBG was measured as the difference between energy intake and expenditure and the results were analyzed in the light of using them to design public health policies and to better understand the weight gain trends in Latin America. Under this approach, some researchers have recently been able to evaluate EIBG in the population of Japan [46] New Zealand [47], and East-Greenland [48]. They found that the EIBG is driven by major environmental, economic, and social factors. Thus, the roles of socio-environmental factors in the body weight status of the adult population are significant indicators, especially if research attempts to assess the impact of public health interventions on different subpopulations based on their sex and BMI.

This study found an association between studied correlates and energy imbalance gaps. Men and subjects of normal weight, overweight and obesity had a higher negative EIBG, compared to women and underweight people, respectively, and differences and similarities between countries were found. It is important to note that previous studies had shown that EIBG increases substantially when BMI moves away from 25 [36,46,47] and is higher with moderate physical activity [49,50].

Fallah-Fini et al. [46,47] also examined the trend in EIBG according to the sex. They found the overall IEBG for women consistently decrease. This outcome is contrary to the results of this study and the experience in American population [36] in which they found that non-Hispanic black and Mexican American women showed a larger positive EIBG than men. There were some significant concerns about women having a positive EIBG. First, obese women during childbearing age could have major health risks, and if they become pregnant there might be some deleterious effects on the future newborn particularly during his early years [51]. Thus, obesity as a result of a positive EIBG in women of fertile age in Latin America can be a risk factor for this segment of the population [52]. However, not only a positive EIBG results in obesity, as observed in this study. The interaction in reaching a sustainable body status over time considers other variables such as the physical activity and body composition of people in a specific environment. In the past, to reach equilibrium in energy balance, an increase in physical activity was necessary, whereas in the modern sedentary society, the equilibrium is obtained by increasing body weight, adapting to the changes in physical activity patterns [1,2].

A heavier body will require more energy to be sustained, particularly if fat-free mass is involved [41]. If energy expenditure was taken from its components (diet-induced energy expenditure, resting energy expenditure and activity-induced energy expenditure), each of these components will be impacted by the energy intake as this could promote changes in energy expenditure as a function of body changes and composition [53]. Therefore, obese individuals would experience metabolic changes as they increase their weight or decrease it. This fact should be taken into account when addressing obesity-reduction strategies [53].

The results of this study showed an important variability among different countries, age brackets, and socioeconomic status, making explicit the difficulties for implementing actions targeting one specific factor. Uauy and Diaz [54], conducted a review of studies about overfeeding in which physical activity was also recorded, and found this variable difficult to assess due to errors in measuring either intake or physical activity. However, Dugas et al. [55], believed that the study of EB and obesity, particularly in low and middle-income countries, are related to poor quality of the diet and unbalanced energy intake.

This was consistent with studies that also reported high interindividual variations and the argued that intra-individual variability was a more important factor to explain this phenomenon. An interesting study conducted by Lewis [56] in a systematic and structured way showed that variations on an individuals’ weight are high and might not be related strictly to intake.

The study of obesity in the past decades has been key for understanding the underlying mechanisms and its effects on health as well as aspects that interact to define the overall body composition of an individual, including environmental, epigenetic and early development factors [57,58,59]. Hence, energy balance studies, while complex, could add valuable information to identifying the actions can be taken from a comprehensive perspective that includes the multiple factors implied in achieving EB.

There is controversy in the methods for assessing EB; Dhurandhar et al. [60] consider that it is unacceptable to still rely on self-reported energy intake or physical activity data to evaluate the energy balance. Other authors defended its use for epidemiological reasons and address larger population groups that otherwise would not be studied through the costly double water method [61,62]. While we are fully aware of these limitations on the methodology, and potential errors they could lead to, the ELANS group agreed on studying the status of energy gaps within the ELANS countries in the most scientific and exhaustive way, thus providing an opportunity for identifying EIBG that are consistent with the rest of the published ELANS findings [63]. In addition, this article is a way to promote the scientific discussion on this controversial matter, which is less addressed in the literature. Finally, the study of energy intake and physical activity seeds a route for further explorations in nutrition epidemiology in Latin America.

### Limitations and Strengths

This study presents some limitations inherent to the methodology selected. First, it is a cross-sectional study which does not allow for establishing causality even with adjustments for covariates. Second, only urban populations were considered, and third, traditional methods for assessing energy balance could not be used. However, the strengths of this study include the large sample size, representative of the urban population of eight countries, the use of two consecutive 24 h recalls, including misreporting as an adjustment for evaluating the energy intake, and the use of validated methods for physical activity assessment.

## 5. Conclusions

This study found that women had a positive energy gap in ELANS countries compared to men. Overweight and obese populations in Argentina, Costa Rica, Ecuador, Peru, and Venezuela showed a significantly lower EIBG than underweight subjects. In addition, this study continues the path for applying methods in nutrition epidemiology that are compatible with large population groups, while being aware of the limitations. These findings confirm the high variability of the EIBG and the correlates that accompany energy intake and expenditure. Systematic research needs to continue in order to make more solid epidemiologic approaches in larger population groups such as ELANS to specifically address interventions in low- and middle-income countries, such as many in Latin America, that help to reduce the prevalence of obesity in addition to eradicating undernutrition.

## Figures and Tables

**Figure 1 ijerph-19-01129-f001:**
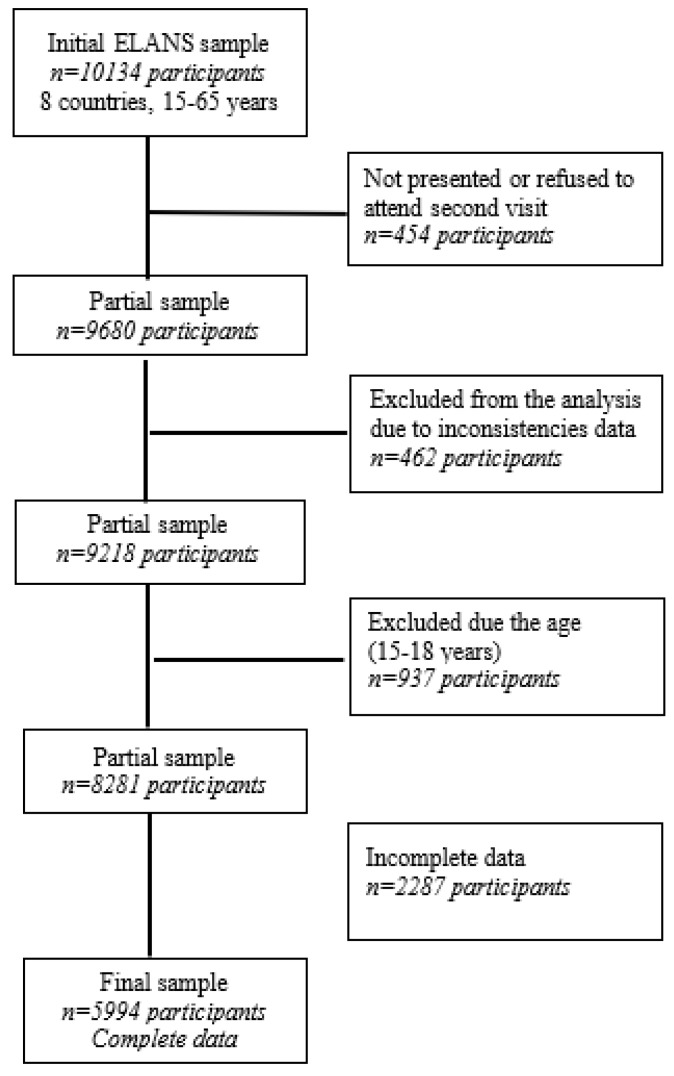
Flow chart of selection of ELANS participants.

**Figure 2 ijerph-19-01129-f002:**
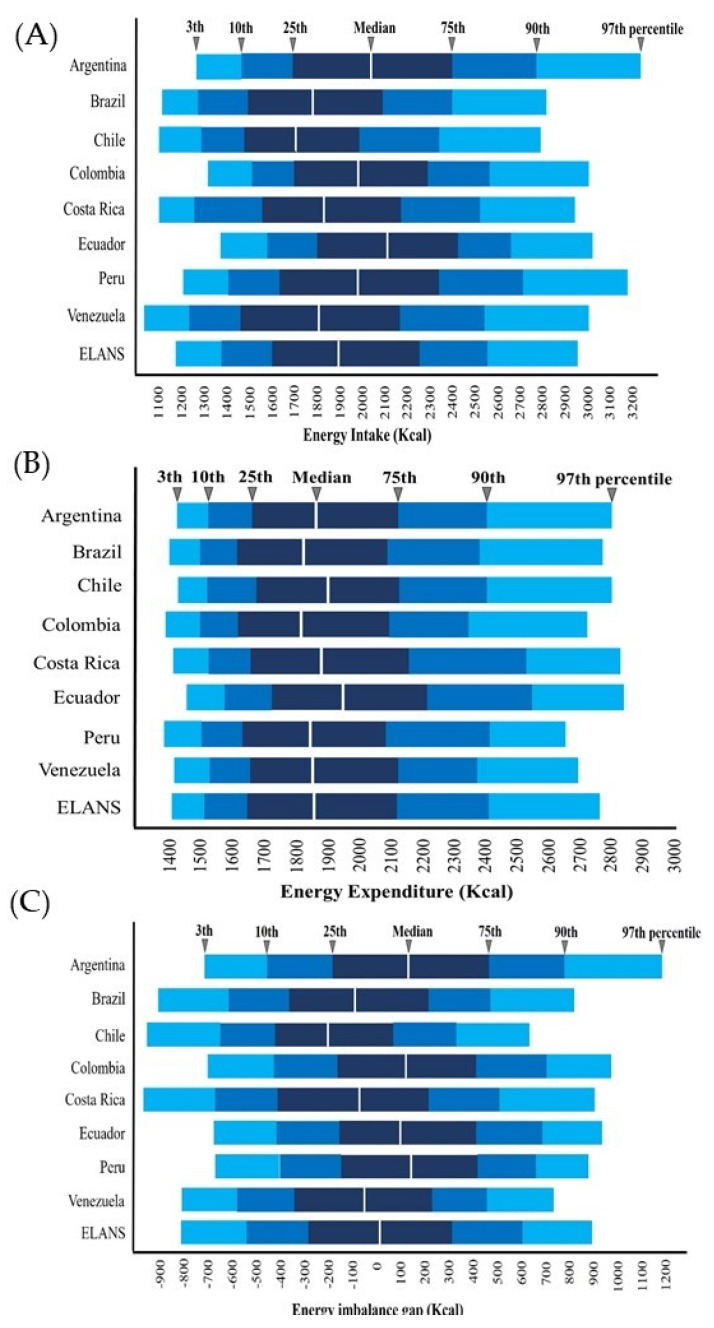
Energy intake (**A**), energy expenditure (**B**), and energy imbalance (**C**) gap percentiles by country.

**Table 1 ijerph-19-01129-t001:** Sociodemographic characteristics by energy intake, energy expenditure and energy balance.

Variables	*N*	%	Mean (95% CI) of EI	Mean (95% CI) of EE	Mean (95% CI) of EB
**Country**					
Argentina	798	13.3	2086.2 (2049.8; 2122.7)	1925.7 (1899.9; 1951.6)	160.5 (127.1; 193.8)
Brazil	1321	22.0	1820.5 (1795.2; 1845.8)	1888.7 (1868.9; 1908.6)	−68.3 (−92.5; −44.1)
Chile	547	9.1	1767.0 (1730.3; 1803.7)	1937.0 (1907.4; 1966.7)	−170.1 (−203.6; −136.5)
Colombia	806	13.4	2020.3 (1989.3; 2051.3)	1884.2 (1859.1; 1909.3)	136.1 (105.2; 167.0)
Costa Rica	495	8.3	1879.1 (1835.2; 1923.1)	1954.2 (1919.4; 1989.0)	−75.0 (−118.1; −32.0)
Ecuador	484	8.1	2128.1 (2089.1; 2167.1)	2005.2 (1972.1; 2038.2)	123.0 (84.6; 161.3)
Peru	771	12.9	2028.3 (1995.8; 2060.7)	1889.6 (1864.9; 1914.3)	138.7 (109.3; 168.0)
Venezuela	772	12.9	1858.2 (1827.3; 1889.0)	1914.3 (1889.9; 1938.8)	−56.2 (−85.5; −26.9)
ELANS	5994	100.0	1939.1 (1926.9; 1951.3)	1915.7 (1906.4; 1924.9)	23.4 (11.9; 35.0)
**Sex**					
Male	2827	47.2	2154.4 (2136.5; 2172.4)	2155.1 (2142.3; 2168.0)	−0.7 (−19.5; 18.1)
Female	3167	52.8	1746.9 (1733.5; 1760.3)	1701.9 (1694.3; 1709.5)	45.0 (30.9; 59.0)
**Age group**					
19 to 34	2669	44.5	2017.3 (1998.4; 2036.2)	1973.5 (1958.4; 1988.7)	43.8 (25.5; 62.1)
35 to 49	1913	31.9	1931.7 (1910.8; 1952.5)	1919.0 (1904.0; 1934.0)	12.6 (−7.2; 32.5)
50 to 65	1412	23.6	1801.4 (1778.6; 1824.2)	1801.8 (1785.7; 1817.8)	−0.4 (−22.6; 21.8)
**Socioeconomic status**					
Low	3113	51.9	1920.1 (1903.1; 1937.1)	1888.7 (1876.4; 1901.0)	31.4 (15.6; 47.2)
Middle	2285	38.1	1956.5 (1937.3; 1975.8)	1939.7 (1924.3; 1955.1)	16.8 (−2.2; 35.9)
High	596	9.9	1971.6 (1931.2; 2012.1)	1964.3 (1932.6; 1996.0)	7.3 (−30.0; 44.7)
Education level					
None and basic	3479	58.0	1923.6 (1907.3; 1939.9)	1899.5 (1887.7; 1911.2)	24.1 (9.0; 39.3)
High school	1897	31.6	1967.1 (1945.8; 1988.3)	1938.7 (1921.7; 1955.7)	28.3 (7.5; 49.1)
Bachelor’s degree	618	10.3	1940.5 (1904.8; 1976.3)	1936.1 (1905.6; 1966.5)	4.5 (−31,5; 40.4)
**Ethnicity**					
Caucasian	2101	35.1	1906.4 (1885.9; 1926.9)	1906.4 (1891.0; 1921.9)	0.0 (−19.8; 19.8)
Mestizo	2766	46.1	1981.0 (1963.2; 1998.9)	1920.1 (1906.7; 1933.5)	61.0 (44.2; 77.7)
Black	378	6.3	1932.8 (1884; 1981.7)	1932.9 (1895.4; 1970.4)	0.0 (−45.8; 45.8)
Indigenous	118	2.0	1905.8 (1825.8; 1985.8)	1890.7 (1831.8; 1949.5)	15.1 (−70.9; 101.1)
Others	631	10.5	1874.3 (1835.8; 1912.7)	1921.6 (1889.9; 1953.2)	−47.3 (−83.0; −11.6)
**Body mass index**					
Underweight	116	1.9	1882.3 (1799.2; 1965.4)	1589.7 (1537.0; 1642.3)	292.6 (216.8; 368.5)
Normal weight	2028	33.8	1950.8 (1930.3; 1971.2)	1805.0 (1790.3; 1819.7)	145.8 (127.2; 164.3)
Overweight	2224	37.1	1945.6 (1925.2; 1966.0)	1934.3 (1919.9; 1948.6)	11.3 (−7.4; 30.0)
Obese	1626	27.1	1919.8 (1896.2; 1943.3)	2051.5 (2033.4; 2069.6)	−131.8 (−153.9; −109.6)

95% CI: confidence interval 95%; EI: energy intake; EE: energy expenditure; EB: energy balance.

**Table 2 ijerph-19-01129-t002:** Multilevel linear regression models for the effect of anthropometric measures, lifestyle, and sociodemographic correlates, on energy intake by country.

Independent Variables	Argentina	Brazil	Chile	Colombia	Costa Rica	Ecuador	Peru	Venezuela	ELANS
**Sex**									
Female ^1^	Reference	Reference	Reference	Reference	Reference	Reference	Reference	Reference	Reference
Male	496.7 (431.9; 561.5)	372.5 (327.2; 417.9)	408.5 (344.4; 472.6)	350.5 (294.2; 406.7)	468.3 (392.8; 543.7)	398.9 (330.4; 467.4)	431.8 (374.4; 489.1)	327.0 (270.3; 383.7)	402.8 (380.9; 424.7)
**Age group**									
50 to 65 ^2^	Reference	Reference	Reference	Reference	Reference	Reference	Reference	Reference	Reference
35 to 49	63.8 (−18.8; 146.3)	149.5 (92.3; 206.7)	97.1 (17.5; 176.7)	135.7 (62.9; 208.6)	137.7 (40.5; 234.9)	131.5 (33.8; 229.2)	140.5 (62.3; 218.8)	55.2 (−23.4; 133.8)	118.6 (89.8; 147.4)
19 to 34	141.4 (55.6; 227.3)	234.9 (173.4; 296.4)	181.9 (94.2; 269.6)	170.7 (97.3; 244.1)	267.6 (162.7; 372.5)	252.7 (157.4; 348.0)	229.2 (150.6; 307.8)	144.7 (67.5; 221.9)	200.5 (171.3; 229.8)
**Socio-economic status**									
Low ^3^	Reference	Reference	Reference	Reference	Reference	Reference	Reference	Reference	Reference
Middle	1.7 (−67.6; 70.9)	13.7 (−38.6; 66.0)	−51.4 (−129.3; 26.5)	48.6 (−14.5; 111.8)	52.7 (−30.3; 135.8)	15.1 (−59.4; 89.5)	−22.4 (−89.4; 44.7)	76.2 (−2.4; 154.8)	17.4 (−6.6; 41.4)
High	−46.7 (−223.1; 129.7)	58.3 (−48.0; 164.7)	−204.3 (−419.0; 10.4)	96.6 (−40.1; 233.3)	146.3 (−3.9; 296.5)	−45.8 (−170.3; 78.7)	−39.2 (−127.9; 49.5)	103.7 (−35.4; 242.8)	35.1 (−8.9; 79.1)
**Education level**									
None and Basic ^4^	Reference	Reference	Reference	Reference	Reference	Reference	Reference	Reference	Reference
High school	−3.9 (−87.8; 80.1)	10.0 (−43.6; 63.6)	−16.1 (−103.1; 70.9)	66.7 (−5.3; 138.7)	8.0 (−103.2; 119.3)	56.4 (−52.9; 165.8)	43.3 (−34.9; 121.5)	−1.1 (−88.9; 86.7)	−6.6 (−32.5; 19.3)
Bachelor’s degree	−133.6 (−307.4; 40.3)	100.9 (−4.0; 205.8)	1.6 (−140.3; 143.6)	59.3 (−34.0; 152.6)	−95.1 (−262.1; 71.9)	45.7 (−114.4; 205.9)	64.8 (−108.7; 238.2)	−8.7 (−84.1; 66.7)	−26.9 (−67.8; 14.0)
**Ethnicity**									
Indigenous ^5^	Reference	Reference	Reference	Reference	Reference	Reference	Reference	Reference	Reference
Caucasian	−179.0 (−416.2; 58.3)	26.4 (−127.1; 179.9)	−166.9 (−436.4; 102.7)	41.9 (−115.8; 199.5)	130.3 (−113.0; 373.5)	82.1 (−218.0; 382.2)	85.6 (−326.1; 497.3)	50.3 (−183.5; 284.1)	4.5 (−76.0; 85.0)
Mestizo	−123.7 (−390.0; 142.7)	−4.6 (−166.4; 157.3)	−108.1 (−378.7; 162.5)	74.4 (−87.8; 236.6)	148.0 (−120.5; 416.6)	125.7 (−90.0; 341.4)	118.3 (−236.1; 472.8)	14.0 (−205.2; 233.1)	58.7 (−19.2; 136.7)
Black		117.9 (−47.4; 283.3)		123.7 (−63.4; 310.7)	−417.0 (−1105.1; 271.2)	−125.3 (−676.9; 426.2)	581.7(−186.1;1349.5)	110.9 (−150.9; 372.7)	6.4 (−84.0; 96.7)
Others	−128.7 (−474.4; 216.9)	62.5 (−109.4; 234.5)	−98.0 (−394.6; 198.7)	−15.1 (−173.3; 143.1)	92.3 (−207.1; 391.7)	103.4 (−313.3; 520.1)	181.1 (−263.0; 625.3)	95.1 (−151.3; 341.6)	−24.6 (−111.8; 62.6)
**Body mass index**									
Underweight ^6^	Reference	Reference	Reference	Reference	Reference	Reference	Reference	Reference	Reference
Normal weight	−73.2 (−314.1; 167.8)	31.8 (−105.0; 168.6)	49.4 (−482.7; 581.4)	220.3 (35.1; 405.4)	−1.4 (−323.3; 320.5)	−51.0 (−296.1; 194.2)	−39.7 (−266.2; 186.7)	−5.1 (−192.1; 182.0)	35.6 (−42.1; 113.2)
Overweight	0.4 (−281.3; 282.0)	59.8 (−88.6; 208.2)	−21.5 (−531.8; 488.8)	203.6 (9.3; 398.0)	−66.7 (−422.7; 289.3)	−67.4 (−309.9; 175.1)	−12.4 (−242.7; 217.9)	25.8 (−174.2; 225.9)	57.3 (−25.0; 139.7)
Obese	53.6 (−217.0; 324.2)	146.2 (3.3; 289.0)	308.7 (−297.7; 915.1)	302.6 (101.9; 503.4)	−4.8 (−300.5; 291.0)	−43.0 (−283.7; 197.7)	124.5 (−137.7; 386.7)	109.1 (−114.9; 333.2)	111.4 (28.6; 194.2)

Adjusted for ^1^ sex, ^2^ age group, ^3^ socio-economic status, ^4^ education level, ^5^ ethnicity, ^6^ body mass index.

**Table 3 ijerph-19-01129-t003:** Multilevel linear regression models for the effect of anthropometric measures, lifestyle, and sociodemographic correlates, on energy expenditure by country.

Independent Variables	Argentina	Brazil	Chile	Colombia	Costa Rica	Ecuador	Peru	Venezuela	ELANS
**Sex**									
Female ^1^	Reference	Reference	Reference	Reference	Reference	Reference	Reference	Reference	Reference
Male	479.6 (444.4; 514.9)	467.4 (441.9; 492.8)	451.0 (410.7; 491.4)	462.0 (428.0; 496.0)	516.3 (469.9; 562.8)	495.6 (448.7; 542.4)	479.0 (445.1; 513.0)	421.7 (390.1; 453.2)	469.4 (457.0; 481.8)
**Age group**									
50 to 65 ^2^	Reference	Reference	Reference	Reference	Reference	Reference	Reference	Reference	Reference
35 to 49	55.1 (11.4; 98.9)	115.6 (83.9; 147.2)	131.1 (82.8; 179.3)	123.3 (81.5; 165.0)	178.7 (119.3; 238.1)	90.4 (32.3; 148.6)	121.1 (76.9; 165.2)	94.1 (55.5; 132.8)	110.9 (95.5; 126.2)
19 to 34	167.6 (121.1; 214.2)	201.8 (165.7; 237.9)	232.4 (175.3; 289.5)	215.2 (170.0; 260.4)	267.4 (202.2; 332.6)	228.8 (160.4; 297.2)	220.9 (173.1; 268.6)	227.3 (182.1; 272.6)	217.3 (200.1; 234.4)
**Socio-economic status**									
Low ^3^	Reference	Reference	Reference	Reference	Reference	Reference	Reference	Reference	Reference
Middle	21.6 (−16.0; 59.2)	17.1 (−12.0; 46.2)	−14.9 (−63.1; 33.3)	50.5 (12.8; 88.1)	5.4 (−47.5; 58.3)	7.9 (−43.1; 58.9)	25.4 (−13.1; 63.9)	−6.5 (−50.9; 37.8)	28.7 (15.3; 42.2)
High	91.6 (−7.0; 190.2)	23.1 (−32.9; 79.1)	−185.0 (−311.2; −58.9)	81.6 (7.5; 155.7)	14.8 (−70.8; 100.4)	41.1 (−42.2; 124.4)	110.9 (59.2; 162.7)	−47.7 (−125.1; 29.7)	53.9 (29.8; 78.0)
**Education level**									
None and Basic ^4^	Reference	Reference	Reference	Reference	Reference	Reference	Reference	Reference	Reference
High school	24.0 (−22.1; 70.0)	31.7 (1.6; 61.7)	52.3 (−0.8; 105.3)	57.0 (14.4; 99.6)	44.3 (−23.7; 112.3)	−29.9 (−103.7; 43.9)	−9.6 (−54.2; 35.0)	49.5 (1.4; 97.7)	1.6 (−12.9; 16.1)
Bachelor’s degree	11.8 (−83.1; 106.8)	37.7 (−18.1; 93.4)	66.7 (−21.4; 154.7)	7.3 (−45.7; 60.3)	117.0 (18.1; 216.0)	25.9 (−86.4; 138.1)	50.7 (−52.6; 153.9)	36.0 (−6.2; 78.1)	10.7 (−12.1; 33.4)
**Ethnicity**									
Indigenous ^5^	Reference	Reference	Reference	Reference	Reference	Reference	Reference	Reference	Reference
Caucasian	16.7 (−109.7; 143.1)	93.1 (4.6; 181.5)	−92.3 (−227.9; 43.3)	−57.9 (−172.0; 56.2)	60.2 (−92.3; 212.8)	71.9 (−132.1; 276.0)	−151.4 (−396.6; 93.9)	24.7 (−91.8; 141.1)	8.8 (−35.0; 52.5)
Mestizo	−7.4 (−128.7; 113.9)	98.9 (5.0; 192.8)	29.4 (−156.4; 215.2)	−72.6 (−160.7; 15.6)	81.6 (−93.5; 256.7)	58.4 (−88.9; 205.8)	−191.7 (−400.6; 17.2)	50.2 (−80.5; 180.9)	12.3 (−32.6; 57.2)
Black		111.4 (25.0; 197.7)		25.3 (−92.6; 143.2)	−165.0(−610.8; 280.9)	211.9 (−74.8; 498.6)	−423.8(−1438.4; 590.9)	59.2 (−103.0; 221.3)	40.4 (−8.3; 89.2)
Others	68.5 (−184.7; 321.7)	104.1 (18.1; 190.1)	−24.8 (−211.3; 161.7)	−18.3 (−134.3; 97.6)	188.9 (12.5; 365.3)	59.6 (−196.0; 315.3)	−121.2 (−511.0; 268.7)	60.9 (−66.7; 188.6)	43.1 (−7.1; 93.2)
**Body mass index**									
Underweight ^6^	Reference	Reference	Reference	Reference	Reference	Reference	Reference	Reference	Reference
Normal weight	136.3 (4.1; 268.5)	184.8 (111.6; 258.0)	304.3 (−77.3; 686.0)	133.7 (25.8; 241.6)	275.1 (78.9; 471.2)	203.4 (41.0; 365.8)	193.2 (39.8; 346.6)	141.6 (30.4; 252.8)	180.3 (135.6; 224.9)
Overweight	300.3 (168.0; 432.6)	330.0 (254.0; 406.0)	442.4 (103.1; 781.6)	335.0 (224.7; 445.2)	447.7 (252.9; 642.4)	361.4 (201.1; 521.7)	334.9 (200.4; 469.5)	550.9 (502.3; 599.5)	342.8 (298.8; 386.8)
Obese	504.2 (340.3; 668.1)	560.2 (468.2; 652.1)	633.5 (322.4; 944.5)	541.6 (404.5; 678.6)	666.1 (459.5; 872.8)	504.7 (326.0; 683.4)	539.5 (410.8; 668.1)	550.9 (432.8; 669.0)	550.9 (502.3; 599.5)

Adjusted for ^1^ sex, ^2^ age group, ^3^ socio-economic status, ^4^ education level, ^5^ ethnicity, ^6^ body mass index.

**Table 4 ijerph-19-01129-t004:** Multilevel linear regression models for the effect of anthropometric measures, lifestyle, and sociodemographic correlates, on energy imbalance gap by country.

Independent Variables	Argentina	Brazil	Chile	Colombia	Costa Rica	Ecuador	Peru	Venezuela	ELANS
**Sex**									
Female ^1^	Reference	Reference	Reference	Reference	Reference	Reference	Reference	Reference	Reference
Male	17.1 (−49.0; 83.1)	−94.8 (−142.0; −47.7)	−42.5 (−108.4; 23.4)	−111.5(−172.3; −50.8)	−48.1 (−129.7; 33.5)	−96.6 (−172.0; −21.2)	−47.3 (−105.6; 11.0)	−94.7 (−150.3; −39.1)	−66.6 (−89.1; −44.1)
**Age group**									
50 to 65 ^2^	Reference	Reference	Reference	Reference	Reference	Reference	Reference	Reference	Reference
35 to 49	8.7 (−74.4; 91.6)	33.9 (−25.8; 93.6)	−34.0 (−116.5; 48.5)	12.4 (−64.5; 89.3)	−41.0 (−147.3; 65.3)	41.1 (−65.2; 147.3)	19.5 (−61.7; 100.7)	−39.0 (−115.9; 38.0)	7.7 (−21.7; 37.2)
19 to 34	−26.2 (−115.6; 63.1)	33.2 (−31.1; 97.4)	−50.5 (−139.8; 38.8)	−44.5 (−124.2; 35.3)	0.2 (−114.4; 114.8)	23.9 (−79.5; 127.3)	8.4 (−71.4; 88.1)	−82.7 (−158.5; −6.8)	−16.7 (−46.9; 13.4)
**Socio-economic status**									
Low ^3^	Reference	Reference	Reference	Reference	Reference	Reference	Reference	Reference	Reference
Middle	−19.9 (−90.3; 50.4)	−3.4 (−58.4; 51.7)	−36.5 (−117.1; 44.0)	−1.8 (−70.0; 66.3)	47.3 (−43.1; 137.7)	7.2 (−75.6; 89.9)	−47.8 (−115.5; 19.9)	82.7 (5.1; 160.4)	−11.3 (−36.1; 13.4)
High	−138.3 (−315.7; 39.1)	35.2 (−71.3; 141.8)	−19.3 (−243.1; 195.6)	15.1(−127.5; 157.6)	131.5 (−24.8; 287.7)	−86.9 (−219.5; 45.7)	−150.1 (−238.6; −61.6)	151.3 (15.1; 287.5)	−18.8 (−62.9; 25.3)
**Education level**									
None and Basic ^4^	Reference	Reference	Reference	Reference	Reference	Reference	Reference	Reference	Reference
High school	−27.9 (−113.3; 57.6)	−21.6 (−77.1; 33.8)	−68.3 (−157.8; 21.1)	9.7 (−68.1; 87.5)	−36.3 (−155.0; 82.5)	86.3 (−33.9; 206.6)	52.9 (−26.6; 132.5)	−50.6 (−136.9; 35.6)	−8.2 (−34.8; 18.3)
Bachelor’s degree	−145.4 (−321.4; 30.6)	63.2 (−44.4; 170.9)	−65.0 (−210.2; 80.2)	52.0 (−47.6; 151.5)	−212.1(−391.3; −32.9)	19.9 (−159.1; 198.9)	14.1 (−157.1; 185.3)	−44.7 (−120.3; 31.0)	−37.6 (−79.4; 4.3)
**Ethnicity**									
Indigenous ^5^	Reference	Reference	Reference	Reference	Reference	Reference	Reference	Reference	Reference
Caucasian	−195.7 (−445.4; 54.1)	−66.6 (−227.8; 94.5)	−74.6 (−356.9; 207.7)	99.8 (−80.7; 280.3)	70.0 (−201.1; 341.1)	10.2 (−322.4; 342.7)	236.9 (−196.9; 670.7)	25.6 (−199.4; 250.6)	−4.3 (−87.3; 78.8)
Mestizo	−116.3 (−376.1; 143.5)	−103.5 (−280.5; 73.6)	−137.5 (−416.5; 141.6)	146.9 (−26.6; 320.5)	66.4 (−218.0; 350.9)	67.3 (−171.2; 305.8)	310.0 (−49.8; 669.8)	−36.3 (−249.2; 176.7)	46.4 (−34.3; 127.2)
Black		6.6 (−163.4; 176.6)		98.4 (−84.0; 280.8)	−252.0 (−975.7; 471.7)	−337.3 (−864.0; 189.5)	1005.5(−234.9; 2245.8)	51.8 (−252.1; 355.7)	−34.1 (−126.3; 58.2)
Others	−197.2 (−536.6; 142.2)	−41.6 (−209.6; 126.4)	−73.2 (−383.0; 236.6)	3.2 (−164.3; 170.7)	−96.6 (−424.5; 231.2)	43.7 (−419.9; 507.4)	302.3 (−179.2; 783.8)	34.2 (−215.9; 284.3)	−67.7 (−154.6; 19.3)
**Body mass index**									
Underweight ^6^	Reference	Reference	Reference	Reference	Reference	Reference	Reference	Reference	Reference
Normal weight	−209.4 (−447.2; 28.3)	−153.0 (−292.5; −13.5)	−255.0 (−828.0; 318.0)	86.6 (−111.2; 284.4)	−276.5 (−649.1; 96.1)	−254.4 (−518.0; 9.3)	−232.9 (−472.3; 6.5)	−146.6 (−327.6; 34.3)	−144.7 (−224.5; −64.8)
Overweight	−299.9 (−587.0; −12.9)	−270.2 (−420.9; −119.5)	−463.9 (−1006.0; 78.3)	−131.3(−338.5; 75.8)	−514.4(−853.5; −175.2)	−428.8(−698.6; −158.9)	−347.3(−579.7; −114.9)	−326.4(−530.8; −122.0)	−285.5 (−369.5; −201.5)
Obese	−450.6 (−729.1; −172.2)	−414.0 (−573.6; −254.4)	−324.7 (−898.7; 249.2)	−238.9(−457.9; −19.9)	−670.9(−1002.9; −338.9)	−547.7(−823.4; −272.0)	−415.0(−663.1; −166.8)	−441.8(−655.5; −228.1)	−439.5 (−525.5; −353.6)

Adjusted for ^1^ sex, ^2^ age group, ^3^ socio-economic status, ^4^ education level, ^5^ ethnicity, ^6^ body mass index.

## Data Availability

The datasets generated and/or analyzed during the current study are not publicly available due the terms of consent/assent to which the participants agreed but are available from the corresponding author on reasonable request. Please contact the corresponding author to discuss availability of data and materials.

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
