# Peer review of "Energy Imbalance Gap, Anthropometric Measures, Lifestyle, and Sociodemographic Correlates in Latin American Adults—Results from the ELANS Study"

_ijerph, 2022, doi:10.3390/ijerph19031129_

Round 1

Reviewer 1 Report

1. Your list of references contains mostly outdated sources; it is recommended that you add references to current research from 2020-2021.
2. The description of table results is insufficiently detailed, there are no conclusions why such or other results and what it is related to.
3. The abstract should contain Novelty /Improvement. 
4. The Introduction is poor. Please explain your research content in this section more precisely. 
5. A flowchart should be added to show the research methodology, to visually present the study better.
6. Please compare the results of your study with previous studies by other researchers and analyze the results completely and novelty.
7. The Conclusion section needs to be described scientifically. Kindly frame it along the following lines:
a) Main findings of the present study
b) Comparison with other studies
c) Implication and explanation of findings
d) Strengths and limitations
e) Recommendation and future direction
8. The scientific novelty of the study is not clear to me. It should be described and substantiated further.
9. In addition, specify how to apply your results in practice? 
10. How was the data for analysis obtained and what materials were used? 
11. What was the period of the study?

Reviewer 2 Report

The manuscript is well written and structured. The subject is interesting and the data is consistent. Very feel grammar mistakes were observed. I have some comments to the authors, as follows.

Title: instead of "correlates", I would suggest changing to "correlations"

Abstract: I suggest including at least one sentence as introduction and explaining why the authors think that this is an important subject

Introduction: there are too many paragraphs that could be merged together, i.e, from line 61 to 73 could be in the same paragraph.

line 61: you should add quotation marks to identify the title of the review that you are talking about

line 74: truncated paragraph

In the introduction, the authors only mention ELAS at the very end and I still don't know what it is. Is it an online survey for all south Americans? It should be explained in the introduction. In the materials and methods the authors can explain how they used the database

In the results, I didn't find in the text anything regarding the figure 1, 2 and 3. They were not mentioned and not explained in the text. In addition, all figures must have a legend that are descriptive enough to be understandable without reading the text. put them together, label them as figure 1A, B and C and explain them in the text/legend.

The discussion section is good, but I have some comments regarding: access to computer to answer the survey (SEL); the ratio of participants of a country over the total population of that country (you can determine which country part. For example, Brazil has 22% of total number of participants, but what is the % of the Brazilian population in the south America?

Reviewer 3 Report

Thank you for allowing me to review the article entitled “Energy imbalance gap, anthropometric measures, lifestyle, and sociodemographic correlates in Latin American adults. Results from the ELANS study ”(ijerph-1444505).

This is a very interesting work in the relationship between nutrition and physical exercise, obesity is really a priority issue due to the trend we see today.

The aimed to study the energy gaps between intake and expenditure of energy in the adult population in Latin America, and its relationships with sociodemographic variables and nutrition status.

Summary:

The abstract could have some introductory phrase, the objective presented is not exactly the same as the one found at the end of the introduction. In the abstract, the abbreviations used must be identified with their meaning, some are there, but others are not, therefore it must be indicated.

The conclusion should be fit for purpose, it should be rewritten.

Key Word: I suggest you add Latin-America.

Introduction:

The introduction should inform about the subject, line 94 I think should be eliminated since the phrase does not make sense ... please check it.

Material and methods

The study design is based on the ELANS study is a household-based multi-national cross-sectional study carried out with the aim of collect reliable and comparable information about energy intake and energy expenditure in representative samples of eight Latin American countries (Argentina, Brazil, Chile, Colombia, Costa Rica, Ecuador, Peru and Venezuela), but the date on which the data was collected must be indicated, as well as whether it is a longitudinal study or a cross-sectional design, as appears from the results presented. It would also be necessary to clarify how the participants in the sample have been selected from the ELANS study. And the date on which it was made.

The methodology used is well described.

Results

There are well presented.

Discussion

In the discussion, it is expected to interpret the results obtained in the context of knowledge, however, its discussion is scarce and lacks this information. It must be expanded.

Conclusions

The conclusions must be derived from the results obtained and presented in the work.

Reviewer 4 Report

The English language needs correction. The text must be made more concise. The methods used are not appropriate for a study of this kind (Dhurandhar NV et al. Energy Balance Measurement: When Something is Not Better than Nothing. Int J Obes (London). 2015:39(7):1109-1113.)

Round 2

Reviewer 1 Report

I am not satisfied with the authors' revisions and explanations.

Author Response

We thank you for the time spent in the review.

Reviewer 3 Report

I have carefully reviewed the new version of the article entitled ”entitled“ Energy imbalance gap, anthropometric measures, lifestyle, and sociodemographic correlates in Latin American adults. Results from the ELANS study ”(ijerph-1444505). As well as the response of the reviewers.

I have verified that the requested clarifications have been introduced, I consider that the article is now more understandable for the readers.

I think the authors have done a great job.

Author Response

(The authors gave the same response as above.)

Reviewer 4 Report

The problems with this manuscript remain.

Author Response

(The authors gave the same response as above.)
